# Sphagnum and Herbaceous Net Ecosystem Exchanges in a Pyrenean Peatland: A Long-Term Study Using the ISBA Model

Raphael Garisoain<sup>1,2</sup>, Christine Delire<sup>1</sup>, Bertrand Decharme<sup>1</sup>, and Laure Gandois<sup>2</sup>

<sup>1</sup>Météo-France, CNRS, Univ. Toulouse, CNRM, Toulouse, France.

<sup>2</sup>CRBE, Universite de Toulouse, CNRS, Toulouse, France

**Correspondence:** Raphael garisoain (raphael.garisoain@univ-tlse3.fr)

**Abstract.** Peatlands play a crucial role in the global carbon cycle, acting as long-term carbon sinks. However, their stability is increasingly threatened by climate change, particularly through rising temperatures and the intensification of droughts. This study focuses on the Bernadouze peatland in the Pyrenees Mountains and aims to validate a newly implemented Sphagnum Plant Functional Type (PFT) in the ISBA land surface model, assess the temporal evolution of carbon fluxes over the past 70 years, and investigate the factors influencing carbon accumulation, with a particular emphasis on drought events.

The model was validated using in situ data, demonstrating reliable carbon flux estimations. Using this validated model, we reconstructed the net ecosystem exchange (NEE) dynamics of the Bernadouze peatland from 1959 to 2022. The results reveal significant interannual variability in NEE, largely driven by air temperature and water table depth. While the peatland has remained a carbon sink, extreme droughts such as those in 1989, 1994, 2003, and most recently 2022 have led to substantial  $CO_2$  emissions.

Our findings suggest that although increasing temperatures have extended the growing season and enhanced gross primary productivity (GPP), the rising frequency and intensity of droughts pose a long-term risk to peatland carbon storage. The dryness index developed in this study appears to be a strong predictor of summer and annual NEE, offering a potential tool for estimating carbon fluxes in peatlands lacking direct measurements.

#### 15 1 Introduction

Peatlands are vital to the global carbon cycle, serving as significant carbon reservoirs and actively exchanging  $CO_2$  and methane with the atmosphere (Gorham et al., 2012). However, the stability of these carbon stocks is increasingly threatened by global warming (Page and Baird, 2016; Carter et al., 2012; Loisel et al., 2021), highlighting the need for precise models to predict their carbon balance. This concern is particularly relevant in mountainous regions, where climate change is expected to be most pronounced (Rogora et al., 2018), potentially affecting the functioning of mountain peatland ecosystems.

Models have been developed to represent the biophysical and biogeochemical processes that occur in peatlands. *Sphagnum* mosses are considered the dominant vegetation in Northern peatlands (Turetsky, 2003) and function differently from vascular plants, lacking stomata to regulate their water content (Clymo and Hayward, 1982). They acquire water either by absorbing atmospheric precipitation or through capillary action directly from the soil surface. Their presence is also crucial for the accu-

mulation of organic carbon in peatlands, as they have a lower litter decomposition rate compared to vascular plants (Scheffer et al., 2001; Lang et al., 2009). As a result, various classes of models have been developed to account for the role of *Sphagnum* mosses, reflecting diverse approaches. Dynamic vegetation and ecosystem models, developed as offline tools without considering atmospheric, climate and carbon feedbacks have incorporated them into their framework (Wania et al., 2009; Frolking et al., 2010; Walker et al., 2017; Gong et al., 2020; Metzger et al., 2016). Furthermore, specific Continental Surface models have been developed for the representation of peatlands (St-Hilaire et al., 2010; Wu and Blodau, 2013b; Wu et al., 2016; Wania et al., 2013; Park et al., 2018). There have also been ongoing efforts to improve the representation of *Sphagnum* mosses in Global Land Surface models, with varying degrees of complexity and scope (Chadburn et al., 2015; Porada et al., 2016; Druel et al., 2017; Qiu et al., 2019; Shi et al., 2015; Grant et al., 2012; Shi et al., 2021).

Efforts to study peatland responses to drought episodes increasingly rely on field measurements and mesocosm experiments (Sterk et al., 2023; Robroek et al., 2024), though research in this area remains limited. A significant limitation of these methods is the scarcity of continuous field measurements capturing medium-term carbon fluxes over decades or longer. Such data would offer critical insights into the frequency and long-term effects of episodic drought events on carbon accumulation. Current carbon flux measurements are generally restricted to contemporary dynamics, which exhibit notable interannual variability, complicating efforts to extrapolate trends over medium or long-term timescales (Young et al., 2021). To explore long-term carbon accumulation dynamics, peat coring analyses provide a reliable means of determining the Average long-term apparent rates of C accumulation (LORCA), offering insights into millennia of accumulation (Turunen, 2003). Additionally, based on the same methodology, the Actual Rate of Carbon Accumulation (ARCA) can be determined, theoretically enabling the observation of carbon accumulation over decades or centuries. Interest in ARCA has grown recently, particularly as a means of assessing the effects of climate and environmental changes on carbon accumulation. However, its reliability has been questioned (Young et al., 2021; Frolking et al., 2014), as ARCA does not accurately reflect the net balance of organic carbon in peatlands. An alternative approach involves modeling exercises that combine peat core age data with empirical models of organic matter decomposition. This allows reconstruction of historical carbon fluxes, linking absorbed and released carbon to initial peatland carbon stock (Yu, 2011; Packalen and Finkelstein, 2014; Frolking et al., 2010; Bunsen and Loisel, 2020; Loisel and Yu, 2013). In this study, we propose another complementary approach, leveraging continental surface models that use atmospheric forcing data as input. Observationally-based atmospheric forcings, available with hourly resolution for some regions since the 1960s, provide a robust basis for modeling peatland evolution over the past several decades. Against the backdrop of Europe's severe drought in 2022 (Faranda et al., 2023), recent research on the Bernadouze peatland in the Pyrenees mountains identified a significant carbon release to the atmosphere during this event, supported by six years of field data (Garisoain et al., 2024). This drought presents a compelling case within a 70-year record, raising questions about its severity as an isolated event versus its representation of a broader trend. Additionally, it prompts examination of whether the peatland's functioning during 2022 aligns with long-term behavior.

To address these questions, this study pursues three key objectives: (1) Implementation and first-site evaluation of a new *Sphagnum* Plant Functional Type (PFT) within the ISBA land surface model. (2) Assessment of the temporal evolution of

https://doi.org/10.5194/egusphere-2025-5248 Preprint. Discussion started: 27 November 2025

© Author(s) 2025. CC BY 4.0 License.

carbon fluxes from the Bernadouze peatland over the past 70 years. (3) Investigation of the factors influencing carbon accumulation, with a particular focus on drought episodes.

#### 2 MATERIALS AND METHODS

# 2.1 Study site

The Bernadouze peatland, located at an altitude of 1,343 m in the eastern French Pyrenees (42.80273°N; 1.42361°E), covers roughly 4.7 hectares and is part of a national biological reserve. Designated as a regional biological reserve since 1983 and a Natura 2000 site since 2007, it is one of the four sites of the French National Peatland Observatory Service SNO-Tourbieres. Characteristic of alpine and southwestern European mountainous peatlands, it is classified as a soligenous fen, continuously receiving water from precipitation and surface runoff. It has formed over the past 5000 years within a 1.4 km watershed with steep slopes averaging 50%. A beech forest surrounds the fen, extending up to 1,800 m. The peatland has an average depth of 2 m, reaching up to 6 m in some areas. The vegetation comprises species typical of both ombrotrophic areas, such as *Sphagnum* palustre and *Sphagnum* capillifolium, and minerotrophic areas, including Carex demissa and Equisetum fluviatile (Henry et al., 2014); see also Figure 1 in Garisoain et al. (2023).

## 2.2 Environmental monitoring

Water table depth (WTD) is monitored by nine automatic piezometer probes (Orpheus Mini water level loggers), with data recorded hourly. As in precedent studies of the same peatland (Garisoain et al., 2023, 2024) we used the S2M reanalysis, which combines the French weather service SAFRAN meteorological reanalysis (8 km resolution) with the SURFEX/ISBA-Crocus snow cover model (Vernay et al., 2022). The S2M chain was applied in a local mode at the scale of the "Couserans" massif, assuming homogeneous weather conditions across the massif for equivalent altitudes but accounting for local topographic features in the calculation of shortwave radiation. The vertical resolution of the S2M model is 300 m, with an hourly frequency, the site's topographical features, including altitude, slope, and aspect of the peatland, were carefully considered. This provides access to all atmospheric variables surrounding the peatland.

#### 2.3 Carbon fluxes

Hourly validated time series of  $CO_2$  fluxes (GPP, ER, and NEE) spanning 2017 to 2022 are available, as detailed by Garisoain et al. (2024). These datasets were obtained through statistical modeling based on monthly  $CO_2$  flux measurements using the static chamber technique. In subsequent sections of this study, these statistical models will serve as a reference.

#### 2.4 Model development

In the present study, we utilize the ISBA model, which is part of the SURFEX land surface modeling platform (Version 9) and serves as the land surface component of the global Earth System Model CNRM-ESM (Delire et al., 2020). ISBA is

widely employed to simulate surface processes, but it does not currently represent mosses or *Sphagnum*, which are essential components of ecosystems such as peatlands. To address this limitation, we introduce a new plant functional type (PFT) designed specifically for modeling mosses and *Sphagnum* within the ISBA framework. This work details the implementation of this novel PFT and the associated modifications to the model to better represent the dynamics of peatland ecosystems. The other PFTs, representing vegetation types such as herbaceous plants and trees, have been thoroughly described in the existing literature (Gibelin et al., 2008).

#### 2.4.1 Photosynthesis of Sphagnum

Plant photosynthesis is modeled according to the description by (Goudriaan, 1986) and (Jacobs, 1994), focusing on the equations involving light dependency (2) and the dependence on mesophyll conductance or  $CO_2$  (1).

$$Am = Am_{max} \times \left(1 - e^{\frac{-gm^*(C_i - \Gamma)}{Am_{max}}}\right) \tag{1}$$

Am is the assimilation rate under light-saturated conditions  $(Kg.m^{-2}.s^{-1})$ ,  $Am_{max}$  is the maximum assimilation rate  $(Kg.m^{-2}.s^{-1})$ ,  $gm^*$  is the mesophyll conductance without water stress  $(m.s^{-1})$ ,  $C_i$  is the internal mesophyll  $CO_2$  concentration (see equation B4) and  $\Gamma$  is the  $CO_2$  compensation point, that's to say the  $CO_2$  concentration below which the plant no longer fixes  $CO_2$ , both in  $Kg.m^{-3}$ .

$$A_n = (Am + R_d) \times \left(1 - e^{\frac{-\epsilon I_a}{Am + R_d}}\right) - R_d \tag{2}$$

 $A_n$  is the net assimilation rate  $Kg.m^{-2}.s^{-1}$ ,  $R_d$  is the "dark respiration"  $Kg.m^{-2}.s^{-1}$ ,  $\epsilon$  the light use efficiency  $Kg.J^{-1}$  et  $I_a$  the photosynthetically active radiation  $W.m^{-2}$ .

$$105 \quad R_d = \frac{Am}{9} \tag{3}$$

is the mitochondrial respiration, empirically fixed.

In ISBA,  $gm^*$  and  $Am_{max}$  depend on the type of PFT (plant type) and the leaf surface temperature following a Q10 function, modified by Collatz et al. (1992) to account for inhibitions (Jacobs et al., 1996). Then,  $gm^*$  is replaced by gm, which allows accounting for plant water stress conditions (see Table A1). Thus, the effect of stomatal closure on photosynthesis in ISBA is controlled by gm, not by gs (the stomatal conductance), which is calculated as a function of  $A_n$  and used solely to determine transpiration fluxes (see equation B5).

To model the photosynthesis of *Sphagnum* mosses, only mesophyll conductance has been considered to contribute to photosynthesis, even though the transfer of  $CO_2$  within the hyaline cells is still poorly understood (Weston et al., 2015). Since

Sphagnum mosses lack stomata, stomatal conductance is no longer defined for the Sphagnum PFT. Therefore, the relationship between Sphagnum moss photosynthesis and Sphagnum moss water content is accounted for through gm. Shi et al. (2021); Walker et al. (2017) described Sphagnum moss photosynthesis using the concept of total Sphagnum moss  $CO_2$  conductance, derived from measurements by Williams and Flanagan (1998) relating Sphagnum moss water content to Sphagnum moss  $CO_2$  conductance. In our case, we relate Sphagnum moss mesophyll conductance and total Sphagnum moss  $CO_2$  conductance. Although they are not the same physical quantity. In Jacobs' description, gm is the closest approximation to total  $CO_2$  conductance. Hence, part of the work involved parameterizing the equation by Williams and Flanagan (1998) to link gm and  $g_{tot}$  according to equation (5), where  $g_{tot}$  is defined as the normalized total Sphagnum moss  $CO_2$  conductance (equation (4) and Figure A1) according to (Gong et al., 2020).

$$125 \quad g_{tot} = \begin{cases} (\beta + \gamma \times w_{Sp} + \eta \times w_{Sp}^2) & \text{if } w_{opt} \ge w_{Sp} \ge w_{min} \\ (1 - 0.8 \times \frac{w_{Sp} - w_{opt}}{w_{max} - w_{opt}}) & \text{if } w_{Sp} \ge w_{opt} \\ 0 & \text{if } w_{Sp} \le w_{min} \end{cases}$$

$$(4)$$

$$gm = \alpha \times g_{tot} \tag{5}$$

$$\eta = \frac{\beta - 1}{w_{opt}^2} \qquad \qquad \gamma = -2 \times w_{opt} \times \eta \tag{6}$$

Where  $w_{Sp}$  is the water content of *Sphagnum* mosses  $(g.g^{-1})$ ,  $g_{tot}$  is the normalized total conductance,  $\beta$ ,  $\gamma$ ,  $\eta$  are coefficients given in Table A2,  $w_{min}$  and  $w_{max}$  respectively the maximum and minimum water content of *Sphagnum* mosses  $(g.g^{-1})$  and  $\alpha$   $(m.s^{-1})$  is a parametrized coefficient given in Table A2.

# 2.4.2 Leaf area index - Canopy scaling

The evolution of Sphagnum moss biomass results from the balance between carbon assimilation through photosynthesis and Sphagnum moss mortality, calculated according to Equation (7):

$$\frac{\Delta B}{\Delta t} = A_c - R_c - \frac{B}{\Delta t} \exp\left(\frac{-t}{\tau}\right) \tag{7}$$

where B is the active biomass of Sphagnum mosses in  $kg.m^{-2}$ ,  $\frac{B}{\Delta t} \exp\left(\frac{-t}{\tau}\right)$  is the mortality of the Sphagnum moss biomass,  $\tau$  is the characteristic mortality time,  $\Delta t$  is a day,  $A_c$  is the net assimilation of the Sphagnum mosses canopy and  $R_c$  is the respiration of Sphagnum mosses canopy ( $A_c = LAI \times A_n$ ,  $R_c = LAI \times R_d$ ).

The LAI is directly calculated from the leaf biomass reservoir B according to:

$$LAI = B \times SLA \tag{8}$$

where SLA (specific leaf area) is a foliar vegetation index representing the leaf area per unit of assimilated carbon ( $m^2 ext{.} m^{-2} ext{.} Kg^{-1}$ ).

$$SLA = e \times N_m + f \tag{9}$$

Here,  $N_m$  is the mass-based nitrogen concentration in the leaf (Calvet and Soussana, 2001). The SLA (constants e and f) is defined for each vegetation type.

Once net assimilation is calculated, multiplying by the LAI allows for scaling from the individual *Sphagnum* strand to the *Sphagnum* canopy. The ISBA model follows the assumption of maintaining constant temperature, humidity, and  $CO_2$  concentration throughout the canopy. Additionally, a radiative transfer model for photosynthetically active radiation was developed by Carrer et al. (2013), allowing for the representation of light diffusion within a canopy based on its LAI. This module accounts for the decrease in light radiation along the vertical profile of the canopy (with more light at the top of the canopy) as well as the diffusion of light towards the lower parts of the canopy. The attenuation of light within the canopy thus leads to a decreasing vertical profile of photosynthesis within the canopy. Leaf respiration, on the other hand, is assumed to be constant along the vertical profile of the canopy. Canopy respiration corresponds to the respiration calculated at the leaf level multiplied by the LAI.

### 2.4.3 Biomass pools

In the original ISBA model, vegetation is represented by up to six biomass reservoirs: leaves, stem, wood, fine and coarse roots, and a small storage pool corresponding to nonstructural carbohydrates (Gibelin et al., 2008; Delire et al., 2020). For grasses/herbaceous, wood and coarse roots are excluded. Leaf biomass evolves based on photosynthetic carbon assimilation and is reduced by turnover, respiration, and allocation to other pools. Leaf area index (LAI) is derived from leaf biomass and specific leaf area, which depends on both plant functional type (PFT) and nitrogen content. Mortality and turnover are PFT-dependent and climate-sensitive, especially for leaves. Photosynthesis and respiration are computed at sub-daily time steps, while the biomass pools evolve on a daily basis.

For Sphagnum, the structure was simplified to reflect its particular growth strategy and morphology. Only two biomass reservoirs are considered: B, representing the photosynthetically active green biomass, and  $B_{brown}$  the dead biomass. The latter is fed by senescence from the green biomass and follows a similar formulation to the original model's decay processes, with an adjusted decay rate specific to Sphagnum. This minimalist representation captures the essential dynamics of Sphagnum growth and decomposition, consistent with its role in peat accumulation and its lack of differentiated organs like leaves or roots.

The brown *Sphagnum* mosses  $(B_{brown})$  are uniformly distributed over 10 cm of soil.

### 2.4.4 Sphagnum water content

To model the evolution of the water content in *Sphagnum* mosses, we followed the work of St-Hilaire et al. (2010), considering a linear relationship between the soil water content at 10 cm and the water content of *Sphagnum* mosses.

$$w_{Sp} = b \times w_{soil,10cm} + c \tag{10}$$

where  $w_{Sp}$  is the water content of *Sphagnum* mosses in  $g.g^{-1}$  and  $w_{soil,10cm}$  is the soil water content at 10 cm in  $m^3.m^{-3}$ . The empirical coefficients b and c are given in Table A2.

The precipitation interception reservoir is considered negligible. Therefore, *Sphagnum* mosses are supplied with water both by capillary action (water available within 10 cm of soil) and by direct interception of precipitation, although the latter is not directly modeled. In the real world, *Sphagnum* mosses obtain water through capillary action and direct interception of rainfall. Here we only represent the effect of capillarity by relating the *Sphagnum* water content to the water content at 10 cm. Precipitation that would have been intercepted by the interception module instead infiltrates directly into the soil, thus modifying the soil water content and consequently the water content of *Sphagnum* mosses.

#### 180 2.4.5 Sphagnum evaporation

Given that *Sphagnum* mosses do not possess stomata, we consider that the latent heat flux from the vegetation is solely due to the evaporation of water from the their epidermal cells. The plant transpiration phenomenon controlled by stomata is therefore eliminated here. Thus, we have:

$$E_{veg} = \frac{F_{veg}}{R_a + R_{sp}} \times (q_{sat}(T_s) - q_a) \times \rho_a \text{ with } R_a = \frac{1}{C_H V_a}$$

$$\tag{11}$$

$E_{veg}$  is the latent heat flux from the vegetation (see equations B5) in  $kg.m^{-2}.s^{-1}$ .  $R_a$  and  $R_{sp}$  represent the air and Sphagnum canopy resistances, respectively.  $F_{veg}$  is the fraction of vegetation,  $q_{sat}(T_s)$  is the specific humidity at saturation at temperature  $T_s$ ,  $T_s$  is the surface temperature and  $q_a$  is the specific humidity of the air at reference altitude  $z_a$ .  $\rho_a$  is the air density at altitude  $z_a$ ,  $C_H$  is the drag coefficient, and  $V_a$  is the wind speed at  $z_a$ .

 $R_{sp}$  has been modeled following the work of Bond-Lamberty et al. (2011) and Gong et al. (2020), who experimentally established a relationship between the resistance of the *Sphagnum* canopy and the water content of the *Sphagnum* mosses. The *Sphagnum* moss resistance to water decreases towards low values when the water content of the *Sphagnum* mosses is high, leading to strong evaporation. Below a threshold value, as the *Sphagnum* mosses dry out, the resistance of the *Sphagnum* mosses increases linearly, allowing to retain a minimal threshold of water in the *Sphagnum* mosses. Here we define:

$$R_{sp} = max(\frac{1}{SWI_{sp}/d}, R_{sp,min}) \tag{12}$$

with d in  $s.m^{-1}$ , d and  $R_{sp,min}$  given in Table A2.

By choosing this approach, we diverge significantly from Bond-Lamberty et al. (2011) and Gong et al. (2020), who link bulk resistivity to the water content of *Sphagnum* mosses. Here, we disregard the water content of *Sphagnum* mosses and instead directly use the soil water content at 10 cm depth through the *Sphagnum* Soil Water Index ( $SWI_{sp}$ ). However, this new

formulation remains consistent with previous ones when representing the *Sphagnum* canopy resistance  $(R_s)$  as a function of the water content of *Sphagnum* mosses (Figure A2).

Following Decharme et al. (2016) to account for soil water stress by using the Soil Wetness Index we define the *Sphagnum* Soil Wetness Index:

$$SWI_{sp} = \sum_{j} \operatorname{sphafrac}_{j} \max(10^{-4}, \min(\frac{w_{soil,j} - w_{wilt,j}}{w_{fc,j} - w_{wilt,j}}, 1))$$

$$\tag{13}$$

Here, sphafrac j is the proportion of brown Sphagnum in layer j,  $w_{soil,j}$  is the soil water content of layer j,  $w_{fc,j}$  is the soil water content at field capacity of layer j, and  $w_{wilt,j}$  is the soil water content at wilting point of layer j, all in  $m^3 ext{.} m^{-3}$ . Beyond 10 cm depth where we assume there are no living Sphagnum anymore,  $SWI_{sp} = 0$ .

 $SWI_{sp}$  varies between  $10^{-4}$  when the soil water content of the *Sphagnum* zone is less than or equal to the wilting point, and 1 when water is not a limiting factor.

The resistance of *Sphagnum* mosses to water is inversely proportional to the availability of water in the surface layers of the soil. The more water there is in the surface layers, the lower the resistance, with a minimum value of  $R_s = 1$ . The resistance then increases according to the inverse function as the water content of the soil decreases.

The water evaporated by the *Sphagnum* mosses is then removed from the top 10 cm of soil, proportional to the layer depth from each layer concerned.

#### 2.4.6 Soil physics

The ISBA model resolves soil heat and water exchanges using a 14-layer scheme over 12 meters depth, minimizing numerical errors in diffusion equations. Thermal depth remains constant at 12 meters, while hydrological depth varies with vegetation. The surface energy balance combines properties of snowpack and soil/vegetation. A 12-layer snow model by Boone and Etchevers (2001) and improved by Decharme et al. (2016) simulates snow properties like energy absorption, density, and melt processes, considering surface albedo and radiation absorption. Heat transfer in soil follows Fourier's law, accounting for soil water content, porosity, and conductivity. Water mass fluxes are described using the Richards equation, incorporating precipitation, snowmelt, freezing/thawing, and vapor transport. Soil hydraulic properties relate to water content and soil texture, with adjustments for ice presence.

#### 2.4.7 Carbon pools in soil

The ISBA model balances plant debris decomposition and microbial activity to represent soil carbon stocks, based on the CENTURY model. Plant debris, including leaves, stems, and roots, is divided into structural and metabolic litter reservoirs, above and belowground. These decompose into three types of soil carbon reservoirs: fast (less than a year), slow (about a decade), and passive (hundreds to thousands of years). This decomposition process drives soil heterotrophic respiration, releasing  $CO_2$  (see Gibelin et al. (2008)).

#### 2.4.8 Thermal and hydraulic properties of peat soils

ISBA calculates thermal and hydraulic properties of soil by combining mineral soil attributes with those of soil organic carbon (SOC), adjusted for the organic matter proportion in each layer. We keep the same method for peat soils. Peat organic matter density is defined using porosity and the density of pure organic matter (1300 kg.m<sup>-3</sup>). The SOC fraction in each layer is derived from databases, with set values of 1. Peat porosity ranges from 0.930 in surface fibric soil to 0.845 in deeper sapric soil, significantly affecting both the thermal conductivity and water retention capabilities of the soil. These variations influence organic matter density and overall soil properties, particularly in peat soils, ensuring accurate modeling of soil thermal and hydraulic dynamics (Decharme et al., 2016).

#### 2.4.9 Biogeochemical processes in peat

The research conducted by Morel et al. (2019) has led to the development of a biogeochemical module capturing various physical and chemical processes occurring within peatlands. This module discretizes the soil into 14 layers, with soil physics resolved for each. It calculates the concentrations of  $O_2$ ,  $CO_2$ , and  $CH_4$  in each layer, enabling representation of biogeochemical processes across the entire vertical profile. Hence, processes such as methanogenesis in anaerobic conditions, methanotrophy (methane oxidation to  $CO_2$  in the presence of  $O_2$ ), and heterotrophic respiration (production of  $CO_2$ ), are described within ISBA. Soil water level and thus O2 concentration in peat regulate these chemical processes. Various equations account for the three different gas transport mechanisms in peat, including transport by plants, ebullition, and diffusion.

The carbon accumulation in the soil and its transfer between layers are represented by an advection term considered constant at  $2mm \cdot yr^{-1}$  (Guenet et al., 2013). The phenomenon of cryoturbation, i.e., the mixing of surface peat layers due to freezing and thawing, is modeled using a diffusion equation following Koven et al. (2009).

#### 2.4.10 Ecosystem respiration

Ecosystem respiration is defined here as the combination of heterotrophic respiration across the peat profile and autotrophic respiration from surface vegetation.

Heterotrophic respiration is calculated for each of the 14 soil layers, based on the  $CO_2$  concentration in each layer. Two sources of  $CO_2$  are modeled for each layer: the production of  $CO_2$  from organic matter decomposition ( $CO_{2,oxic}$ ) and the production of  $CO_2$  resulting from methane oxidation ( $CO_{2,methane}$ ), indirectly linked to methane concentration in each layer and thus to methanogenesis.

$$255 \quad CO_{2,oxic}^i(z) = \theta(z)k_i(z,Tg)C_i(z) \times min(1,\frac{C_{max}(z)}{\Delta t} \frac{1}{\sum_j \theta(z)k_j(z,Tg)C_j(z)}) \tag{14}$$

Here, i and j represent the types of carbon reservoirs (metabolic and structural litter, active C, slow C, passive C).  $CO_{2,oxic}$  production is determined by the organic matter decomposition rate  $(\theta(z)k_i(z,Tg))$ , which initially depends on soil temperature and moisture content Morel et al. (2019). The function  $\theta(z)$ , varying between 0.05 (dry soil) and 1 (above field capacity),

accounts for soil moisture's impact on microbial activity. In this study, we didn't find a significant reduction of organic matter decomposition due to soil dryness. Hence we decided to remove this dependence. The decomposition rate  $(k_i(Tg))$  follows an Arrhenius-like equation (Q10), accounting for increased decomposition with temperature rise. The last part of the equation considers oxygen availability, potentially limiting reaction.  $C_{max}(z)$  represents the maximum carbon mass producible from available oxygen.

Methane oxidation  $CH_4 + 2O_2 \rightarrow CO_2 + 2H_2O$  contributes to  $CO_2$  production, especially in deeper layers. For litter above ground,  $CO_2$  is directly released to the atmosphere. In deeper layers, diffusion, gas transport by plants (PMT), and evapotranspiration facilitate  $CO_2$  escape. Gas diffusion in soil layers leads to vertical gas movements based on concentration gradients. PMT depends on leaf area index (LAI), while evapotranspiration is influenced by vegetation type.

#### 2.5 Water table depth diagnosis and dryness index development

First, based on the ISBA outputs, we derived Equation 15, which relates, for each soil layer, the change in volumetric water content to the change in water equivalent height, assuming the layer is saturated.

$$w_{j,sat}(t)\frac{\partial h_j(t)}{\partial t} = d_j \frac{\partial w_{j,soil}(t)}{\partial t}$$
(15)

with  $w_{j,sat}$  the volumetric water content of soil layer j at saturation,  $h_j$  the height of water in the layer j,  $d_j$  the vertical width of the layer j,  $w_{j,soil}$  the volumetric water content of soil layer j.

Then, we derived the ISBA-diagnosed WTD by summing over the soil layers down to 2 m depth, i.e. those where variations in water content are significant. This calculation assumes that the variation of  $w_{sat}$  with depth z is negligible, allowing it to be factored out of the summation. (see Figure A3 (a)):

$$WTD_{ISBA} = \sum_{j} \Delta h_j = \sum_{j} \frac{\Delta w_{j,soil} \times d_j}{w_{j,sat}}$$
(16)

where  $\Delta h_j$ , and  $\Delta w_{j,soil}$  represent the changes in water equivalent height and soil volumetric water content, respectively, calculated explicitly through the temporal discretization of Equation 15.

The dryness index is based on the work of (Garisoain et al., 2024) (see Figure A3 (b)):

$$dif(t) = Tair_{normalized}(t) - WTD_{normalized}(t)$$

$$(17)$$

We use daily mean of Tair(t) and WTD(t) to calculate dif(t) Normalization of each variable (X) was done following  $X_{normalized} = \frac{X - X_{min}}{X_{max} - X_{min}}$ . A minimum WTD value of 1m was imposed to elevate the normalized WTD values, since the diagnosed WTD tends to produce water tables that do not drop sufficiently during droughts. This adjustment allows the normalized WTD to better discriminate drought periods, which would otherwise be overly smoothed by the index (see Figure A3.(b)).

$$f(t) = \begin{cases} \operatorname{dif}(t), & \text{if } \operatorname{dif}(t) > 0\\ 0, & \text{otherwise} \end{cases}$$
 (18)

$$DI = \int_{july}^{august} f(t) dt \tag{19}$$

#### 290 2.6 Statistical analyses

To compute correlation coefficients, trends and associated p-values, we used the statsmodels Python library. In particular, the Ordinary Least Squares (OLS) (https://www.statsmodels.org/dev/generated/statsmodels.regression.linear\_model.OLS.html) regression function was employed to fit a linear model and perform an F-test to assess the statistical significance of the relationship.

To assess the relative contribution of each season to interannual variability in net ecosystem exchange (NEE) (Figure 6), we applied SHAP (SHapley Additive exPlanations), a game-theoretic approach widely used for interpreting machine learning models. Specifically, we trained an Ordinary least squares Linear Regression model from sklearn python library (https://scikit-learn.org/stable/modules/generated/sklearn.linear\_model.LinearRegression.html to predict annual NEE from seasonal values (summer, autumn, spring, and winter). We then used SHAP values to estimate how much each seasonal predictor contributed to the model output. Rather than relying on model coefficients or explained variance, we computed the mean absolute SHAP values across all observations to quantify each season's average influence. These contributions were normalized to express their relative importance as percentages. This approach provides a robust and interpretable measure of feature relevance, even for collinear predictors. The SHAP methodology is described in detail at: https://shap.readthedocs.io/en/latest/index.html.

A simple linear model using  $R^2$  or an F-test tends to attribute explanatory power to the first variables entered into the model. When predictors are correlated as is the case here with seasonal NEE components (e.g., summer NEE, autumn NEE) this can lead to shared variance being unfairly credited to one variable over another. In other words, correlated variables may "steal" importance from each other, depending on the order in which they are introduced. In contrast, SHAP values, based on Shapley values from cooperative game theory, average the contribution of each variable across all possible combinations of input features. This results in a fair and consistent allocation of importance, even when variables are highly interdependent or collinear.

#### 2.7 Experimental protocol

At the Bernadouze site, we obtained a 64-year meteorological data series (1959-2022) from the S2M reanalysis, which was used as input for the ISBA land surface model. To establish a contemporary carbon balance for the various carbon compartments, we simulated a 7000-year spin-up to account for the peatland's age, repeating the 64-year atmospheric forcing. This approach ensured a realistic accumulation of carbon in the soil reservoirs. The model was run in a one-dimensional configuration, at a single grid point corresponding to the location of the Bernadouze peatland.

For model validation (section 3), we constrast simulations with 100% *Sphagnum* cover and 100% herbaceous cover. Herbaceous cover was simulated with the use of the ISBA PFT boreal grassland (BOGD). For the remainder of the study (section 4), the vegetation distribution is assumed to consist of 70% herbaceous plants and 30% *Sphagnum* mosses, based on the cartography provided by Henry et al. (2014).. Accordingly, the GPP, ER, and NEE values presented correspond to this vegetation mix. A subsection is dedicated to analyzing the sensitivity of carbon fluxes to vegetation composition.

#### 325 3 Model validation over the 2017-2023 period

# 3.0.1 carbon fluxes

Figure 1. Daily (a) gross primary productivity and (b) ecosystem respiration fluxes compared between statistical models (grey) (Garisoain et al., 2024) and ISBA over the 2017-2022 period for herbaceous (blue) and Sphagnum mosses (orange).

Figure 1 compares three time series from 2017 to 2023, illustrating Gross Primary Productivity (GPP), Ecosystem Respiration (ER) and Net Ecosystem Exchange (NEE) modeled by a statistical model (grey) and the ISBA model for *Sphagnum* (orange) and Herbaceous (blue). The top plot (a) shows GPP with  $R^2$  values of 0.6 and 0.62 (RMSE of 1.9 and 1.6  $\mu$ mol. $^{-2}$ . $s^{-1}$ , see Figure A4 (a), (b)), indicating moderate agreement between the models. The middle plot (b) shows ER with  $R^2$  values of 0.82 and 0.44 (RMSE of 0.9 and 1.6  $\mu$ mol. $^{-2}$ . $s^{-1}$ , Figure A4 (c), (d)), revealing stronger agreement for ER, particularly for *Sphagnum* respiration. The bottom plot (c) shows NEE with  $R^2$  values of 0.1 and 0.2 (RMSE of 1.6 and 1.3  $\mu$ mol. $^{-2}$ . $s^{-1}$  Figure A4 (e), (f)). The ISBA model captures the interannual variability of GPP, ER and NEE effectively (Figure A5), Consistently showing slightly higher values at the beginning of the growing season, much higher values at the end of the growing season, and stronger winter ecosystem respiration (ER) compared to the statistical model. Herbaceous vegetation is more sensitive to summer droughts than *Sphagnum*, leading to reduced GPP and ER in summer, especially during the 2022 drought, which caused a sharp decline in herbaceous GPP and ER.

# 3.0.2 Water table depth

Figure 2. Hourly in situ mean water table depth is shown in grey, with its standard deviation in shaded areas, compared to the hourly diagnosed water table depth of ISBA over the 2017-2022 period for two different types of vegetation herbaceous (blue) and Sphagnum mosses (orange).

Figure 2 compares the mean water table depth (WTD) from in situ measurements (grey with shaded standard deviation) to the ISBA diagnosted WTD (orange for *Sphagnum* mosses and blue for herbaceous) over the period 2017-2022. The ISBA model generally follows the observed data, capturing the overall trends and seasonal variations (Figure A5) in WTD although there are occasional deviations, particularly at the end of the growing season and during drought events where the observed data shows higher variability. Overall, the ISBA model demonstrates rather satisfactory performance in simulating WTD ( $R^2 = 0.47$ , RMSE = 0.09m for herbaceous and  $R^2 = 0.33$ , RMSE = 0.1m for *Sphagnum* Figure A6). The simulated WTD with herbaceous plants is consistently lower than with *Sphagnum* mosses.

# 4 Results: Water and carbon balance of the peatland over the last 60 years

#### 4.1 Environmental variables

Figure 3. (a) Mean and maximum annual air temperature  $^{\circ}$  C, (b) annual cumulate precipitation  $mm.year^{-1}$  from the S2M reanalysis and (c) mean water table depth m diagnosed from ISBA outputs, along with their trends as red dashed lines and corresponding p-values.

Figure 3 consists of three panels, labeled (a), (b), and (c), illustrating the evolution of mean and maximum annual temperature, annual cumulative precipitation, and annual mean water table depth (diagnosed from ISBA) from 1959 to 2022, along with their respective trends and p-values. Annual mean temperature (a) shows a significative (p-value 

Annual cumulative precipitation (b) ranges from approximately 1000 mm to 2250 mm, while the Water Table Depth (WTD) (c) varies between -0.16 and -0.29 m over the whole period. A steady increase in the level of the water table is observed from 1967 to 1983, followed by a sharp decline, reaching its minimum in 1989. After this, the WTD rises again and stabilizes, although notable interannual fluctuations persist until the end of the period. Interestingly, the fluctuations in annual precipitation closely mirror those of the mean annual WTD, suggesting a strong correlation between these two variables. The trends of annual cumulative precipitation and annual mean WTD move in opposite directions, with precipitation increasing and the level of the water table decreasing. However, both trends are not significant (p-values > 0.05).

# 4.2 Net ecosystem exchanges

# 4.2.1 An overall carbon sink despite strong inter-annual variability

Figure 4. (a) Annual net ecosystem exchange  $(gC.m^{-2}.year-1)$  from 1959 to 2022. (b) Hourly cumulated net ecosystem exchange  $(gC.m^{-2})$  from 1959 to 2022.

Figure 4 illustrates (a) the annual cumulated Net Ecosystem Exchange (NEE) and (b) the hourly cumulated NEE from 1959 to 2022. The lowest annual NEE, -171  $gC.m^{-2}.year^{-1}$ , is modelled in 2011, while the highest, 122  $gC.m^{-2}.year^{-1}$  in 2022.

Panel (b) reveals a steady decline in cumulative NEE over the study period, indicating that the ecosystem acts as a net carbon sink by absorbing more carbon than it emits. From 1959 to around 1990, the decline is gradual, but after 1990, the rate accelerates, reflecting increased carbon sequestration. This trend persists through the 2000s and 2010s, accompanied by greater year-to-year variability.

Despite annual fluctuations, the overall trend highlights a subtle but consistent accumulation of carbon by vegetation over time.

# 4.2.2 Seasonality of GPP, ER and NEE over the 1959-2022 period

Figure 5. Seasonal evolution of cumulated (a) minus gross primary productivity, (b) ecosystem respiration, (c) net ecosystem exchanges from ISBA over several time periods: 1959-1980 in blue, 1980-2001 in orange, 2001-2022 in green with interannual variability represented as a 90% confidence interval. Superimposed, the 2022 NEE seasonality simulated by ISBA (red curve) and the statistical model (purple curve).

Figure 5 depicts the seasonal mean and interannual variability in gross primary production (a), ecosystem respiration (b), and net ecosystem exchange (c) over several time periods, specifically 1959-1980 (blue), 1980-2001 (orange), and 2001-2022 (green), with a focus on the year 2022 both from ISBA (red line) and the statistical model (purple line). Each subplot shows

cumulative values with their respective mean and interannual variability at 90% confidence interval. In panel (a), GPP values are negative, indicating carbon uptake by vegetation, with more pronounced uptake during the growing season (April to September). Panel (b) shows ER values, which are positive, representing carbon release. Respiration increases during warmer months, with higher rates, suggesting intensified ecosystem respiration. Panel (c) combines GPP and ER to show NEE. The GPP intensifies over time and becomes increasingly pronounced across the three periods. The same trend is observed for respiration. As a combination of these two variables, NEE exhibits greater seasonal variability, which is not necessarily easy to grasp from this graph but is more apparent in Figure 6. The mean NEE values for the periods 1980-2001 and 2001-2022 show a similar trend, except during winter, when the NEE of the 2001-2022 period shifts towards positive values. Nonetheless, over the annual cycle, both of these periods display more negative NEE values compared to the 1959-1980 period.

The seasonality of GPP (a) and ER (b) has gradually intensified over time, as observed in the three periods. For the NEE (c), the trend is less straightforward, but its seasonality has become increasingly pronounced. Additionally, the duration of the growing season has extended, as reflected by the earlier spring and later summer inflection points of cumulative NEE (solid blue, orange and green curves (c)).

Focusing on 2022, the NEE curve (red) deviates entirely from the interannual variability of the three periods combined, starting in July and continuing through November. Comparing this with the statistical model for 2022 (purple curve), we observe that despite differences in seasonality between ISBA and the statistical model, both agree in highlighting 2022 as a year significantly outside the confidence intervals. While GPP and ER for 2022 remain within the 2001–2022 interannual variability, the GPP is notably close to the lower limit (-GPP to the upper limit), underscoring the exceptional nature of this year in terms of carbon flux dynamics.

Figure 6. Seasonal contributions to annual NEE across four time periods: 1959-1980 in blue, 1980-2001 in orange, 2001-2022 in green, 1959-2022 in grey. Each bar represents the relative importance of a season in explaining the total NEE, as determined by Shapley regression coefficients.

Figure 6 shows that across all time periods, summer is the season that contributes the most to determining the annual NEE, with a contribution of 48% over the 1959-2022 period. Autumn and spring follow, alternating in second place depending on the time period, with contributions of approximately 21% and 19%, respectively, over the entire period. Finally, winter has the lowest contribution, around 11% for the total period.

For the period 2001-2022, we observe a decrease in the summer contribution, notably at the expense of spring, followed by autumn, and finally winter. This result is consistent with the observations in Figure 6 regarding the extension of the growing season. However, the summer period remains the primary driver of the annual NEE evolution. Therefore, we have decided to focus on studying the drivers of summer NEE in the remainder of the paper, while keeping in mind the influence of other seasons, which seem to seem to become increasingly important with rising temperatures.

# 4.3 Joint Influence of Air Temperature and Water Table Depth on Summer NEE Variability

Figure 7. (a) Summer net ecosystem exchanges (bold colormap) compared to the annual one (shaded colormap) from 1959 to 2022. (b) Dryness Index from 1959 to 2022 with its trend as red dashed line and associated p-value.

Panel (a) of Figure 7 shows the evolution of cumulative summer NEE (red) compared to cumulative annual NEE (blue) from 1959 to 2022. The cumulative summer NEE tends to "drive" the cumulative annual NEE ( $R^2 = 0.71$  Figure A7 or contribution of 48% Figure 6), almost always sharing the same sign, except in years near equilibrium (cumulative annual NEE = 0).

Panel (b) illustrates the evolution of the dryness index from 1959 to 2022. The trend shows an increasing pattern, with p = 0.054, which is slightly above the conventional significance threshold of 0.05. As the dryness index increases, the cumulative summer NEE also increases, showing a good correlation ( $R^2 = 0.6$ , Figure A8). The dryness index exceeded 6 in only one year from 1959 to 1980, in five years from 1980 to 2001, and in six years from 2001 to 2022. This suggests an increase in the frequency of high dryness index episodes, indicating a rising frequency of summer droughts. Four years also stand out with particularly high dryness index values: 1989, 1994, 2003, and 2022, all of which occurred after the 1980s.

#### 5 Discussions

#### 5.1 Model validation and vegetation sensitivity

The primary objective of this paper was to evaluate the newly implemented Sphagnum plant functional type (PFT) in the ISBA land surface model. A specific parameterization was developed to represent the relationship between the photosynthetic activity of Sphagnum mosses and their water content. In this framework, Sphagnum water content is linearly related to the soil water content in the top 10 cm. The concept of total  $CO_2$  conductance in Sphagnum was also incorporated, along with a model for Sphagnum water evaporation, in which the resistance to water loss from the moss layer depends on the soil moisture in the same top layer. The Sphagnum PFT was validated using available in situ data from the Bernadouze peatland. Simulations of carbon fluxes (photosynthesis and respiration) showed good agreement with observations ( $R^2 = 0.6$  and  $R^2 = 0.82$ , respectively). Validation of the water cycle was performed indirectly by comparing model outputs with water table height measurements. However, a more detailed validation of the water cycle remains necessary and would require a dedicated study. Eddy covariance data would be essential to directly assess the parameterization of evaporation from the Sphagnum layer. Similarly, extending the evaluation across multiple sites would help refine the parameter values and assess their transferability, although this was beyond the scope of the present study. The aim here was not to optimize parameter values across environments, but to assess whether a realistic site-scale behavior could be reproduced using values drawn from the literature and applied at a single, well-instrumented site.

The model for herbaceous plants is based on a parameterization that has already been validated globally in the past (Gibelin et al., 2008). This study compares ISBA model data with the results of statistical models described in Garisoain et al. (2024), which themselves have uncertainties. As for the net ecosystem exchange (NEE), since it results from the modeling of two distinct processes (GPP and ER), it accumulates the errors present in both contributions. Moreover, the NEE values in question are relatively low in intensity, as they are close to a carbon cycle equilibrium state, making small prediction errors even more significant. This explains the difficulty in obtaining high R-squared values when predicting NEE.

Figure 8. Probability density function of annual cumulated net ecosystem exchange over 1959-2022. In black, the vegetation mix corresponds to 70% herbaceous and 30% Sphagnum. In orange a 100% Sphagnum mix and in blue a 100 % herbaceous mix. In shaded areas, the 95% confidence intervals corresponding to the variation of the vegetation mix in the form  $\alpha \times NEE_{Sphagnum} + \beta \times NEE_{herbaceous}$  with  $\beta = 1 - \alpha$  and  $\alpha$  varying from 0 to 1 in steps of 0.01.

Figure 8 highlights the changes in the distribution of the annual cumulated Net Ecosystem Exchange (NEE) over 1959–2022. The black curves represent the probability density of NEE for a vegetation mix of 70% herbaceous and 30% *Sphagnum*, with shaded regions representing the 95% confidence intervals accounting for variations in vegetation composition.

We observe that while the overall shape of the annual cumulative NEE probability density remains largely unchanged, significant differences arise depending on the vegetation type, particularly around the peak of the distributions. *Sphagnum* mosses amplify NEE extremes, either enhancing  $CO_2$  absorption by vegetation or increasing  $CO_2$  release to the atmosphere, resulting in a flatter distribution compared to herbaceous vegetation. In contrast, herbaceous plants have a buffering effect, with a distribution more concentrated around the main peak and a secondary, smaller peak slightly below +100  $gC.m^{-2}.year^{-1}$ , resulting in a bimodal pattern. These findings highlight the importance of considering different vegetation types, as they respond differently to changing environmental conditions.

To go further, the idea would now be to develop an interactive vegetation model between *Sphagnum* mosses and herbaceous, rather than considering them separately with a purely linear interaction, and thus couple their carbon and water cycles (Verma,

1996). This approach would also make it possible to render the vegetation partition between *Sphagnum* mosses and herbaceous dynamic over time, by including, for example, competition processes between these two types of plants (Lippmann et al., 2023; Heijmans et al., 2008; Wu and Blodau, 2013a; Gong et al., 2020).

# 5.2 The key predictors of annual net ecosystem exchange (NEE)

The methodology developed in this study aimed to investigate the variability and evolution of carbon fluxes in the Bernadouze peatland from 1959 to 2022 using a continental land surface model validated for the present period (2017–2022). This approach provides insight into the long-term functioning of the peatland on a century-scale timescale, which has been scarcely explored in the literature due to the lack of suitable tools. This novel methodology provides access to an unprecedented temporal scale, enabling current observations to be interpreted within a broader historical perspective.

Over the past 64 years, the Bernadouze peatland has shown marked variability in net ecosystem exchange (NEE), while overall maintaining its role as a carbon sink. This variability is strongly influenced by climatic and hydrological conditions, particularly precipitation, water table dynamics, and air temperature, as highlighted in previous research (Yurova et al., 2007; Laine et al., 2019). The reconstruction of water table height, together with the development of a dryness index that integrates both air temperature and water table depth, offers a robust explanation for the observed fluctuations in carbon fluxes, as also supported by other studies (Helbig et al., 2022). VPD (Vapor Pressure Deficit) is also an important factor to consider, particularly in relation to vegetation development, as it influences both GPP and plant transpiration. In Bernadouze, VPD is low and exhibits little variation (Figure A9 (a)), suggesting that it does not have a strong impact on the resulting NEE flux. Additionally, VPD is strongly correlated with temperature, a factor that thus captures much of its influence. A recent study (Chen et al., 2023) has highlighted that in northern hemisphere peatland ecosystems, VPD has a neutral effect on vegetation and does not necessarily lead to the closure of stomata in vascular plants. The humid environmental conditions, along with the presence of bryophytes, help meet the growing atmospheric demand for water vapor. Air temperature and water table depth remain the primary factors explaining the variability in NEE.

It is also observed that, in Bernadouze, summer NEE is the dominant contributor to the annual carbon balance, though the relative influence of other seasons varies across time periods. In recent decades, transitional seasons such as spring and autumn have become increasingly significant compared to earlier years. Understanding how climate change influences NEE seasonality offers key insight into the complex dynamics of carbon fluxes and the shifting balance between source and sink processes throughout the year, as also emphasized by Helbig et al. (2022). The dryness index developed in this study also appears to be a good proxy for summer NEE and, consequently, for annual NEE (Figure A8 and A7). On other peatland sites where carbon flux measurements are not available, this index could potentially serve as a preliminary source of information to estimate the carbon balance of peatlands.

#### 5.3 Climate change and droughts episode

Over the past 64 years, the Bernadouze peatland has experienced an increasing frequency of severe droughts, as indicated by the calculated dryness index (Figure 7 (b) and Figure A3 (b)). These events have contributed to the destabilization of the NEE balance, particularly in 2022. Both the ISBA land surface model and the statistical model by Garisoain et al. (2024) despite some differences in seasonal representation agree that from July to November 2022, conditions fell completely outside the range of interannual variability. Similar dry summers have occurred in the past, notably in 1989, 1994, and 2003, and have consistently led to significant  $CO_2$  emissions into the atmosphere. The years 1989 and 2003 are recognized as having experienced different types of drought conditions in France, ranging from multi-year precipitation deficits (1989–1990) to short, hot, and dry periods (2003) (Vidal et al., 2010). Similarly, 1994 is also identified as a year with a hot summer, preceded by a winter precipitation deficit in Southern Europe (Vautard et al., 2007). The dryness index effectively captures these hot and dry summers, which impact vegetation, its development, and consequently, the NEE flux.

As the growing season lengthens and GPP increases due to rising air temperatures, a compensatory effect appears to be at play. The peatland's greening and higher summer GPP fluxes currently help mitigate the impact of droughts, allowing it to remain a carbon sink. Over the 2001-2022 period, spring and autumn have played a growing role in shaping annual NEE, suggesting that these transitional seasons, along with winter, may become increasingly influential in the future, potentially counterbalancing summer carbon losses. However, the longevity of this balance remains uncertain. Similar studies on European forest ecosystems have also identified the 2022 drought as an exceptionally severe event, with significant consequences for vegetation and, in particular, the carbon sink function of forests. Driven by climate change, such episodes are expected to become more frequent and intense in the future (van der Woude et al., 2023). Over the long term, worsening drought conditions could eventually outweigh these compensatory mechanisms, potentially shifting peatlands from carbon sinks to carbon sources.

The significant increase in annual maximum temperatures about 8°C over 64 years raises concerns about the vegetation's ability to withstand such extreme warming. Studies on potential shifts in plant composition under climate change could provide valuable insights into the future of these ecosystems (Antala et al., 2022; Dieleman et al., 2015). This further emphasizes the need to integrate a broader range of plant communities and their interactions into land surface models, to more accurately represent ecosystem dynamics and their role in the carbon cycle.

#### 6 Conclusion

This study highlights the importance of accurately representing Sphagnum mosses in land surface models to simulate peatland carbon dynamics under changing climatic conditions. Validation of the new Sphagnum PFT within the ISBA model demonstrated its ability to reproduce observed carbon fluxes with reasonable accuracy. Analysis of the Bernadouze peatland over the past 64 years revealed that while it has remained a net carbon sink, increasing drought frequency and severity, particularly

exemplified by the 2022 event, are destabilizing its carbon balance. The findings emphasize the critical role of vegetation composition, hydrological conditions, and seasonal climate dynamics in modulating peatland carbon fluxes. They also suggest that although compensatory mechanisms currently maintain peatland sink function, future intensification of droughts driven by climate change could potentially shift these ecosystems from carbon sinks to carbon sources. This underscores the urgent need to integrate interactive vegetation dynamics and drought responses into land surface models to better project peatland contributions to the global carbon cycle under future climate scenarios.

Code and data availability. The data presented in this study are available at https://doi.org/10.5281/zenodo.16984992 (Garisoain et al., 2025).

The S2M dataset is freely accessible via the AERIS data center at https://doi.org/10.25326/37#v2020 (Vernay et al., 2023). The S2M data are provided by Météo-France, CNRS, CNRM, and the Centre d'Études de la Neige through AERIS.

The model used in this study is open-source. The ISBA model, as implemented in this work, is part of SURFEX version 9 and can be downloaded from the SURFEX platform: http://www.umr-cnrm.fr/surfex/.

Competing interests. The contact author has declared that none of the authors has any competing interests.

Acknowledgements. The authors gratefully acknowledge Matthieu Lafaysse for providing the S2M data and for his valuable support.

Author contributions. RG, CD, BD and LG conceptualized and designed the study. RG modified and implemented the model, conducted formal analysis of the results, and led the writing of the original draft with contributions from all co-authors. All authors participated in reviewing, editing, and finalizing the manuscript.

# Appendix A: Figures and Tables

Figure A1. Normalised total Sphagnum conductance as a function of Sphagnum water content

Figure A2. Sphagnum canopy water resistance as a function of Sphagnum water content

Figure A3. (a) Diagnosed water table depth (WTD) from 1959 to 2022 for the Sphagnum-herbaceous vegetation mix. (b) Light blue: normalized WTD, red: normalized air temperature (Tair), pink: normalized Tair minus normalized WTD, dark blue area under the curve represents the dryness index (DI). Normalization of each variable (X) was done following  $X_{normalized} = \frac{X - X_{min}}{X_{max} - X_{min}}$ . WTD<sub>min</sub> = 1m is taken from observations and not the diagnosed one from ISBA.

Figure A4. Comparison of daily ecosystem photosynthesis and respiration from the statistical model with: (a) the new Sphagnum photosynthesis parameterization, (b) the new Sphagnum ecosystem respiration, (c) the previous herbaceous photosynthesis, (d) the previous herbaceous ecosystem respiration.

Figure A5. Daily annual cycle of (a) Gross Primary Productivity, (b) Ecosystem Respiration, (c) Net Ecosystem Exchange, (d) Water Table Depth from the statistical model in black, the ISBA sphagnum model in orange and the ISBA herbaceous model in blue.

Figure A6. Houlry ISBA-diagnosed water table depth (WTD) with herbaceous vegetation as the dominant cover is compared to hourly in situ WTD in the left panel, while the right panel presents ISBA-diagnosed WTD with Sphagnum as the dominant vegetation versus in situ WTD.

Figure A7. Annual NEE versus Summer NEE for each year from 1959 to 2022.

Figure A8. Summer NEE versus Dryness Index for each year from 1959 to 2022.

Figure A9. (a) Annual mean of vapor pressure deficit (VPD) (Pa) and scatter plot between VPD and air temperature; (b) Annual mean of relative humidity (%) from 1959 to 2022 derived from the S2M reanalysis.

Table A1. Changes between Sphagnum and C3 herbaceous plant functionnal type in ISBA

| Parameters        | Units                                               | C3 Herbaceous                                                                              | Sphagnum                                      | Comments                                            |
|-------------------|-----------------------------------------------------|--------------------------------------------------------------------------------------------|-----------------------------------------------|-----------------------------------------------------|
| $\Gamma(25)$      | ppmv                                                | 45                                                                                         | 45                                            |                                                     |
| $\epsilon_0$      | $Kg_{CO_2}.J^{-1}.PAR$                              | $0.017 \times 10^{-6}$                                                                     | $0.017 \times 10^{-6}$                        |                                                     |
| $Am_{max}(25)$    | mg.m <sup>-2</sup> .s <sup>-1</sup>                 | 1.7                                                                                        | 1.0                                           | Jacob's hypothesis: $Am_{max} = Vc_{max}/2$ .       |
| f                 | (-)                                                 | $f = f_0^* (1 - \frac{D_s}{D_{max}^*}) + f_{\min} \frac{D_s}{D_{max}^*}$                   | $f = f_0^*$                                   | f does not vary with air humidity                   |
| $f_0^*$           | (-)                                                 | $f_0^* = 0.95$                                                                             | $f_0^* = 0.8$                                 |                                                     |
| $g_m$             | m.s <sup>-1</sup>                                   | $g_m^* = e^{2.381 - 0.6103 \ln(D_{max}^*)}$                                                | $g_m = \alpha g_{\text{tot}}$ (Eq. 5)         | SWI = 1                                             |
|                   |                                                     | $g_m = e^{2.381 - 0.6103 \ln(D_{max})}$                                                    | $g_m = \alpha g_{\text{tot}}$ (Eq. 5)         | $SWI_c < SWI < 1$                                   |
|                   |                                                     | $g_m = g_{m, 	ext{minimum}}$                                                               | $g_m = \alpha g_{\text{tot}} \text{ (Eq. 5)}$ | $SWI \leq SWI_c$                                    |
|                   | $Kg_{H_2O}.Kg_{air}^{-1}$                           | $D_{max}^* = 0.045$                                                                        | Non used                                      | SWI = 1                                             |
| $D_{max}$         |                                                     | $D_{max} = D_{max}^{X} + $ $(D_{max}^{*} - D_{max}^{X}) \frac{SWI - SWI_{c}}{1 - SWI_{c}}$ | Non used                                      | $SWI_c < SWI < 1$                                   |
|                   |                                                     | $D_{max} = D_{max}^{X} \frac{SWI}{SWI_{c}}$                                                | Non used                                      | $SWI \leq SWI_c$                                    |
| au                | Days                                                | 150                                                                                        | 80                                            |                                                     |
| SLA               | m <sup>2</sup> .Kg <sup>-1</sup> <sub>DryMass</sub> | $e = 5.56; f = 6.73; N_m = 1.3$                                                            | $e = 12; f = 5; N_m = 1.3$                    |                                                     |
| Root zone         | _                                                   | 20 cm, exponential pro-<br>file distribution                                               | no roots                                      | Uniform distribution of Sphagnum over the top 10 cm |
| Vegetation height | m                                                   | $\frac{LAI}{6}$                                                                            | 0.05                                          | Impacts Drag coefficient (H and LE fluxes)          |

**Note 1:**  $SWI_c = 0.3 \text{ m}^3 \text{.m}^{-3}$ 

Note 2:  $g_m^*, D_{max}^*$  are the same quantities as  $g_m, D_{max}$  but without hydric stress

Note 3:  $D_{max}^X = 0.4$ , this is the maximum value of  $D_{max}$ . Note 4: The calibration of the above relationships between gm and  $D_{max}$  was carried out with gm expressed in  $mm.s^{-1}$  and Dmax in  $g.Kg^{-1}$ 

| Parameter    | Value                                  | Equation |
|--------------|----------------------------------------|----------|
| α            | $0.0004  (\text{mol}^{-1}.\text{m}^3)$ | 5        |
| β            | -0.5                                   | 4        |
| $\gamma$     | 0.5                                    | 4        |
| $\eta$       | -0.0416                                | 4        |
| b            | $27.6  (\mathrm{g.g^{-1}})$            | 10       |
| c            | $-0.5  (\mathrm{g.g^{-1}})$            | 10       |
| d            | $1 (s.m^{-1})$                         | 12       |
| $R_{s,min}$  | $1e^{-4} m.s^{-1}$                     | 12       |
| $w_{\min}$   | $2.4 (g.g^{-1})$                       | 4        |
| $w_{ m max}$ | $22 (g.g^{-1})$                        | 4        |
| $w_{ m opt}$ | $6(g.g^{-1})$                          | 4        |

Table A2. Parameters used for the Sphagnum PFT and their associated equations.

# Appendix B: ISBA equations

$$Am_{max}(Ts) = \frac{Am_{max}(25) \times Q_{10}^{(Ts-25)/10}}{[1 + exp[0.3(T1 - Ts)][1 + exp[0.3(Ts - T2)]]}$$
(B1)

where  $Am_{max}(25)$  is  $Am_{max}$  at  $25^{\circ}C$ ,  $Q_{10}$  is fixed at 2.0, Ts is the skin temperature in  $^{\circ}C$  and T1 and T2 are reference temperature values. gm in unstressed soil moisture conditions,  $gm^*$ , depends on temperature via the same  $Q_{10}$  function as  $Am_{max}$ .

$$\Gamma(Ts) = \Gamma(25) \times Q_{10}^{(Ts-25)/10}$$
 (B2)

 $\Gamma(25)$  is  $\Gamma$  at  $25^{\circ}C$ . Here  $Q_{10}$  is fixed at 1.5.

$$\epsilon = \epsilon_0 \left( \frac{Ci - \Gamma}{Ci - 2\Gamma} \right) \tag{B3}$$

$\epsilon$  is the initial quantum use efficiency, where  $\epsilon_0$  is the maximum quantum use efficiency.

$$C_i = fC_a + \Gamma(1 - f) \tag{B4}$$

The internal  $CO_2$  concentration  $C_i$ , is directly derived from the  $CO_2$  concentration in the air  $C_a$  and from f which is detailed in table A1.

$$E_{g} = (1 - F_{veg})\rho_{a}C_{H}Va(\alpha q_{sat}(T_{s}) - q_{a})$$

$$E_{v} = F_{veg}\rho_{a}(\frac{\delta}{R_{a}})(q_{sat}(T_{s}) - q_{a})$$

$$E_{tr} = F_{veg}\rho_{a}(\frac{1 - \delta}{R_{c} + R_{c}})(q_{sat}(T_{s}) - q_{a})$$
(B5)

The evaporation of the vegetated surface is the sum of the evaporation of the soil  $(E_g)$  and the evaporation of the vegetation  $(E_{veg})$ :  $E = E_g + E_{veg}$ . The evaporation of the vegetation is itself distributed between the direct evaporation  $(E_v)$  due to the fraction of folliage covered by water intercepted and the transpiration  $(E_{tr})$ :  $E_{veg} = E_v + E_{tr}$ .

 $F_{veg}$  is the fraction of vegetation,  $\delta$  the fraction of folliage covered by intercepted water and  $R_s$  is the canopy resistance taking into account the upscalling of the cuticular and stomatal resistance.

For the modelling of Sphagnum,  $\delta$  is set to 0, and  $R_s$  is changed in  $R_{sp}$  (in this context, historical transpiration effectively corresponds to the evaporation from Sphagnum).

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
