# Peer review of "Sphagnum and Herbaceous Net Ecosystem Exchanges in a Pyrenean Peatland: A Long-Term Study Using the ISBA Model"

_EGUsphere, 2025_

## Referee Comment (RC1)

In their manuscript titled "Sphagnum and Herbaceous Net Ecosystem Exchanges in a Pyrenean Peatland: A Long-Term Study Using the ISBA Model" the authors present their parameterization of a Sphagnum plant functional type for $CO_2$ flux modelling using the ISBA land surface model, evaluate the extended model through comparison with a statistical model based on in-situ measurements (2017 – 2022), and use the extended model to reconstruct long-term (1959 - 2022) $CO_2$ flux and water table dynamics. Interannual variability in summer NEE is then related to a drought index developed by the authors.

I agree with the authors that considering the role of *Sphagnum* when modelling peatland greenhouse gas fluxes in highly important as *Sphagnum* moss is among others highly relevant for peat formation. Similarly, investigating the influence of droughts on the carbon sink function of peatlands is highly topical considering the increase in frequency and severity of drought events.

Despite the high relevance of the topics addressed in the manuscript, I have a few major comments that I recommend to being addressed prior to potential publication:

First, I think that all three key objectives of the study would benefit for more detailed analyses. For this, you could consider splitting the study into two manuscripts – one focussing on the evaluation of the new *Sphagnum* PFT and one on the long-term reconstruction of NEE dynamics and the identification of their environmental controls.

Regarding the first research objective, I have some concerns about the model validation. As you mention yourself in the discussion section of the manuscript, the model results are validated against the results of a statistical model that is based on monthly in-situ data obtained from chamber measurements. First, it would be helpful to include some information about the spatial distribution/replication of chamber measurements in the manuscripts to give an indication of their spatial representativeness (even thought this information is given in Garisoain et al., 2024). Second, while six years is a comparatively long time series of validation data, I am concerned about the low temporal resolution (monthly) of the in-situ data and the consequential use of a statistical model to reconstruct daily flux dynamics for validation of the IBSA model. I agree that it would be highly beneficial to instead or additionally use an eddy covariance data set or a chamber data set with higher temporal resolution for the model validation.

Furthermore, I think that for model validation the in-situ data (the statistical model results) should also be compared to the ISBA model results using the actual vegetation distribution of the peatland (30% *Sphagnum*, 70% herbaceous) instead of just 100% *Sphagnum* and 100% herbaceous species.

From your model validation it seems that the *Sphagnum* PFT improves model performance (alignment between IBSA results and in-situ data) mainly for ecosystem respiration (ll. 330-332). From my point of view this is a key result that requires a paragraph in the discussion section relating this finding to the Sphagnum PFT parameterization.

As an idea for future refinements of the *Sphagnum* PFT parameterization: As you already consider $CH_4$ oxidation as a source of $CO_2$, it would be interesting to add the (enhancing) effect of a *Sphagnum* layer on $CH_4$ oxidation (e.g. Larmola et al., 2010; https://doi.org/10.1890/09-1343.1).

Regarding the second and third research objectives on long term NEE dynamics and their drivers, you hint at the growing importance of shoulder season fluxes with a lengthening growing season. I think that this finding would be worth investigating further as in-situ measurements in northern peatlands and consequently many studies are often limited to the growing season.

Minor comments:

I do not think that there is a need to always repeat all the information that is given in the figure captions also in the main text. Instead, I would recommend to just refer to the Figure in brackets at the end of a sentence.

In the Materials and Methods chapter it would be helpful to clarify further which ones of the described model equations you altered and which ones were kept as they are. As the manuscript and especially the methods chapter is quite long you could consider leaving out sections 2.4.6 to 2.4.10 as the described process parameterizations were already included in the original model (if I am not mistaken) and instead refer to a respective description of the original model.

l. 262: What do you mean with "reaction"? Maybe "decomposition"?

Figure 3: Which vegetation distribution was used for these model simulations? 30% *Sphagnum* and 70% herbaceous?

l. 350 "a) shows a significant […]

l. 371 consistent and increasing accumulation, right? Maybe this could be worth mentioning.

Figure 5: I think that the seasonality in NEE, GPP, and ER would be easier to see here if they were not displayed as cumulative fluxes. Only ll. 388-390 refer to the cumulative representation – Figure 5 could therefore be moved to the appendix and be replaced by the respective annual time series.

Figure 6: Please add a definition of the seasons here.

l. 408: Remove one "seem to"

Figure 7 (caption): I don't understand what you mean by "bold colormap" and "shaded colormap".

I think that Figure 8 together with ll. 443-453 could be moved from the Discussion to the Results chapter.

The Discussion chapter could be strengthened by more specifically discussing the key findings of the study. In it's current form a large part of the discussion refers to shortcomings of the study and future research objectives.

ll. 473-480: I don't quite understand the introduction of VPD here. First you state that VPD is important to consider (without a reference) but in the next sentences you show that VPD does not have a significant effect on NEE at your research site or in northern peatlands in general. Maybe rephrasing this paragraph can clarify your point.

Figures A4-A6: It would be interesting to add another column of plots comparing the results of the statistical model with the ones from the ISBA model assuming 30% *Sphagnum* and 70% herbaceous species.

Figure A5: Please add to the figure captions which years are included in the figures.